

# Terminal motions of Longbasaba Glacier and their mass
# contributions to proglacial lake volume during 1988–
# 2018
Junfeng Wei[1], Shiyin Liu[2,3], Te Zhang[1], Xin Wang[1], Yong Zhang[1], Zongli Jiang[1],
Kunpeng Wu[2,3], Zhen Zhang[4]
[1]School of Resource & Environment and Safety Engineering, Hunan University of Science and
Technology, Xiangtan, 411201, China
[2]Institute of International Rivers and Eco-Security, Yunnan University, Kunming, 650091, China
[3]State Key Laboratory of Cryospheric Sciences, Northwest Institute of Eco-Environment and
Resources, Chinese Academy of Sciences, Lanzhou, 730000, China
[4]School of Geomatics, Anhui University of Science and Technology, Huainan, 232001, China
*Correspondence to*: Junfeng Wei (weijunfeng@hnust.edu.cn)
**Abstract.** The interaction between a glacier and its glacial lake plays an increasingly important role in
glacier shrinkage and proglacial lake expansion, and it increases the risk of glacial lake outburst floods
(GLOFs). Longbasaba Glacier is directly contacted by a moraine-dammed lake with a high outburst risk
in the central Himalayas, and has drawn a great deal of attention from scientists and local governments.
Based on Landsat images and *in-situ* measurements, the evolution records of the shrinkage of
Longbasaba Glacier and the corresponding expansion of its proglacial lake were determined for
1988–2018, and the mass contributions of glacier shrinkage to the increase in lake water volume were
assessed. During the past three decades, Longbasaba Glacier has experienced a continuous and
accelerating recession in glacier area and length but accompanied by the decelerating surface lowing and
ice flow. Consequently, Longbasaba Lake has expanded significantly at an accelerating rate. The glacier
surface lowering played a predominant role in the mass contribution of glacier shrinkage to the increase
in lake water volume, while ice avalanches were the main potential trigger for failure of moraine dams
and subsequent GLOF events. Due to the areal expansion, decreasing mass contributions from parent
glacier shrinkage, and some mitigation measures by local governments to improve the drainage systems,
the potential risk of outburst for Longbasaba Lake has continuously decreased during the last decade.
## 1 Introduction
Responding to climate warming during recent decades, the main mountain ranges across the world have
exhibited continuous and accelerating glacier shrinkage (Zemp et al., 2015; Brun et al., 2017; Yang et al.,
2019). The rapid reduction of mountain glaciers plays an increasingly important role in both the areal
and water volume expansion of glacial lakes (Zhang et al., 2015; Song et al., 2017; Zhang et al., 2017;
Yang et al., 2018), and subsequently, has increased the potential risk and destructiveness of glacial lake
outburst floods (GLOFs) (Wang et al., 2016; Nie et al., 2017). For lake-terminated glaciers with debris-
covered tongue, the mass/energy interactions of the thermal regime and ice avalanches between the
glacier front and the lake water result in rapid glacier wastage, which creates subsequent proglacial lake
expansion and is prone to causing an accelerated reduction of the parent glaciers (Carrivick and Tweed,



2013; Fujita and Sakai, 2014). Eventually, lake-terminated glaciers exhibit more significant shrinkage,
which provides a significant mass budget and increases the risk of GLOFs (Emmer, 2017; Zhang et al.,
2019). GLOFs and their accompanying debris flows have become the predominant glacial hazard and
cause ruinous impact on downstream ecosystems, communities, infrastructure, and economic
developments (Fujita et al., 2008; Nie et al., 2018; Zhang et al., 2019). In the Third Pole, the majority of
glacial lakes develop in the Himalayan range, and have experienced an overall areal expansion of ~14%
from 1990 to 2015 (Zhang et al., 2015; Nie et al., 2017). In this region, potentially dangerous glacial
lakes are widely distributed (Wang et al., 2012, 2015), and more than 70 GLOF events have been reported
(Khanal et al., 2015; Veh et al., 2018). Approximately 80% of the reported GLOFs were initiated by an
abundant ice mass suddenly entering a proglacial lake due to ice avalanches on the glacier terminal (Awal
et al., 2010; Nie et al., 2018).
Longbasaba Lake is a typical potentially dangerous glacial lake with a higher outburst risk in the
Himalayas (Wang et al., 2015; Wang et al., 2018). A lake outburst in this location would significantly
threaten the livelihoods and activities of the local people in downstream countries, for example, the
transportation/communication facilities and hydropower stations (Yao et al., 2012). Hence, the evolution
of Longbasaba Glacier/Lake and their mass interactions are necessary in order to assess the prediction of
GLOF events and gain the attention of scientists and local government departments (Wang et al., 2008;
Yao et al., 2012; Nie et al., 2017; Wang et al., 2018). In this study, we aim to assess the mass contribution
of glacier shrinkage to the increase in lake water volume by monitoring the motions of the glacier
terminal and subsequently to extract the ratios of the mass budgets contributed by the ice flow/retreat of
the glacier terminal and the changes in the glacier surface elevations. Finally, the retreat patterns of the
parent glacier and their impacts on the proglacial lake are discussed.

## 2 Study area

Longbasaba Glacier is contacted by a moraine-dammed lake and located at the source area of the Pumqu
River on the northern slope of the central Himalayas (Fig. 1a). This glacier covered an area of 28.4 $km^2$
in 2010, with a length of 8.7 km and a debris-covered area of 1.06 $km^2$ on the tongue (3.7%) (Guo et al.,
2015). There are numerous serac clusters and small supraglacial lakes on the glacier tongue (Fig. 1b).
Many crevasses have formed in the glacier front and commonly cause ice avalanches, which causes ice
floe masses of various sizes over the surface of the proglacial lake (Fig. 1c). Longbasaba Lake is in direct
contact with the parent glacier and remains at a high risk of outburst (Wang et al., 2016). The proglacial
lake had an area of 1.22 $km^2$ in 2009, with a maximum length of 2.210 km from east to west and a
maximum width of 0.685 km from south to north (Yao et al., 2012). According to the *in-situ*
measurements taken using an echo sounder in 2009, the water level of Longbasaba Lake was 5499 m
and the average and maximum depths were 48 m and 102 m, respectively, with a water volume of 0.064
$km^3$ in 2009 (Yao et al., 2012). The glacier front retreated by 1264 m (40 m $a^{-1}$) from 1977 to 2009, which
resulted in a glacial lake expansion of 223%. In addition, an accelerating recession of the glacier terminal
(63.8 m $a^{-1}$) was observed from 2005 to 2009, with a rapid areal expansion rate of 0.040 $km^2$ $a^{-1}$ for
Longbasaba Lake (Wang et al., 2016).

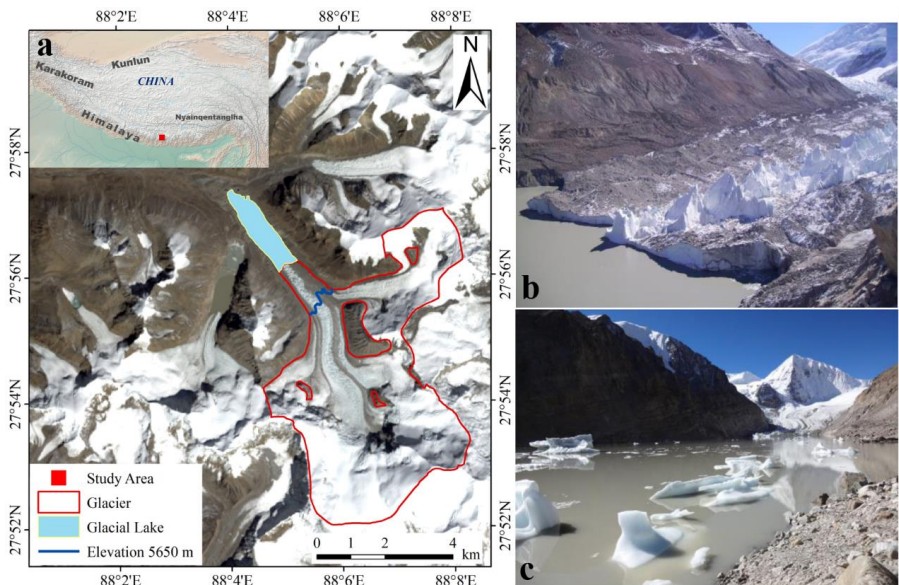

**Figure 1. (a)** The study area. The outlines of Longbasaba Glacier/Lake were detected from the Landsat OLI image taken in 17 October 2018. The background of the eagle-eye map was available from Natural Earth. The blue line shows the location of the ice fall. **(b)** Crevasses, serac clusters and debris cover occurred over the glacier tongue. Some ice avalanches could be found at the flank of the glacier terminal. **(c)** Ice floes widely distributed over the lake surface.

## 3 Data and methods

### 3.1 Glacier reduction and lake expansion

The outlines of Longbasaba Glacier/Lake during the 1988–2018 period were manually generated from Landsat TM\ETM+\OLI images using pan-sharpening employing principle-component analysis. These multispectral, multitemporal images are available for free from the United States Geological Survey (USGS). They are orthorectified with the SRTM DEM and ground-control points from the Global Land Survey 2005 (GLS2005), with a spatial resolution of 30 m and a WGS1984/EGM1996 coordination system (Woodcock et al., 2008). The horizontal accuracies of the Landsat images are better than one pixel to each other or to non-differential GPS data (Bolch et al., 2010; Guo et al., 2015).

In total, 34 Landsat TM\ETM+\OLI images acquired during the investigation period were used in this study (Tab. 1). The Landsat images covering the region of interest were dramatically affected by frequent snow and cloud cover. Then, we preferentially choose the Landsat images acquired in September and October without snow and cloud cover. Other high-quality images acquired in the adjacent months (e.g., July and August) were used to detect the precise outlines of the glacier and lake when there was no perfect image from September or October. In addition, scanline errors (SLC-off scenes acquired by the ETM+ sensor since early summer 2003) also created obstacles in generating the glacier/lake outlines. Then the SLC-off images were used only to define the positions of the glacier front.





**Table 1.** Landsat images utilized to detect the terminal retreat and ice flow of Longbasaba Glacier. Images[*] with
heavy cloud cover or scanline errors (SLC-off scenes) were just for the position generation of the glacier front.

| Date | Image ID | Sensor | Date | Image ID | Sensor |
|------|----------|--------|------|----------|--------|
| 1988/09/12 | LT51390411988256BKT00 | TM | 2003/11/25 | LT51390412003329BKT00 | TM |
| 1989/09/23 | LT41390411989266XXX01 | TM | 2004/10/10 | LT51390412004284BKT00 | TM |
| 1990/06/14 | LT51390411990165BKT00 | TM | 2005/10/13 | LT51390412005286BKT00 | TM |
| 1991/09/21 | LT51390411991264BKT00 | TM | 2006/10/16 | LT51390412006289BKT00 | TM |
| 1992/09/23 | LT51390411992267BKT00 | TM | 2007/10/03 | LT51390412007276BKT01 | TM |
| 1993/10/12 | LT51390411993285BKT00 | TM | 2008/10/21 | LT51390412008295BKT00 | TM |
| 1994/09/29 | LT51390411994272ISP00 | TM | 2009/09/22 | LT51390412009265KHC00 | TM |
| 1995/04/09 | LT51390411995099BKT01 | TM | 2010/06/21 | LT51390412010172KHC00 | TM |
| 1996/10/20 | LT51390411996294ISP00 | TM | 2011/06/08 | LT51390412011159BKT00 | TM |
| 1997/07/03 | LT51390411997184BKT01 | TM | 2013/12/22 | LC81390412013356LGN01 | OLI |
| 1998/10/10 | LT51390411998283BKT00 | TM | 2014/10/06 | LC81390412014279LGN01 | OLI |
| 1999/05/22 | LT51390411999142BKT00 | TM | 2015/10/09 | LC81390412015282LGN01 | OLI |
| 2000/10/15 | LT51390412000289BKT01 | ETM+ | 2016/10/11 | LC81390412016285LGN01 | OLI |
| 2001/10/26 | LE71390412001299SGS00 | ETM+ | 2017/10/30 | LC81390412017303LGN00 | OLI |
| 2002/10/29 | LE71390412002302SGS00 | ETM+ | 2018/10/17 | LC81390412018290LGN00 | OLI |
| 1995/07/30 | LT51390411995211BKT00[*] | TM | 2012/10/08 | LE71390412012282PFS00[*] | ETM+ |
| 2010/10/03 | LE71390412010276SGS00[*] | ETM+ | 2013/10/11 | LE71390412013284SG100[*] | ETM+ |


Subsequently, the areal variations in Longbasaba Glacier/Lake during the investigation period were
generated, and the main flowlines were extracted to assess the changes in the glacier length. By
combining the variations in the position and shape of the glacier front during specific periods, the patterns
of the terminal motions were assessed, including terminal retreat and ice avalanches. The changes in the
surface elevation of Longbasaba Glacier were extracted from the High Mountain Asia Gridded Glacier
Thickness Changes from Multi-sensor DEMs, Version 1 (HMA_Glacier_dH) during two subsequent time
periods of 1975−2000 and 2000−2016 (Maurer et al., 2018). This data was extract based on a series of
stereo scenes from KH-9 HEXAGON in 1975 and ASTER data acquired from 2000 to 2016 by fitting
robust linear trends. The data used is available for free from National Snow and Ice Data Center (NSIDC),
with a horizontal resolution of 30 m. The asserted accuracy of the full data is $\pm$ 0.42 m a$^{-1}$ as derived
from the non-glacier terrain (Maurer et al., 2018). Nevertheless, we obtained a higher accuracy of $\pm$ 0.04
m a$^{-1}$ for the two investigation periods using the method of Burn et al. (2017).
**3.2 Characteristics of glacier surface velocity**
The Landsat images described above were also used to extract the glacier surface velocity field from
image pairs based on cross-correlation feature tracking processing using the free software module Co-
registration of Optically Sensed Images and Correlation (COSI-Corr) (Leprince et al., 2007; Gantayat et
al., 2014; Ruiz et al., 2015; Ayoub et al., 2017). A co-registered image pair containing two Landsat images
was iteratively cross-correlated on sliding windows. Finally, two horizontal ground offset fields





(East\West and North\South) and a signal-to-noise ratio (SNR) were calculated for each pixel. The SNR
value reflects the quality of the registration. The surface velocity of an individual pixel was subsequently
generated by combining two horizontal offsets with a higher SNR threshold of >0.95.
The mean surface velocity of the glacier (MSVG) was extracted over the glacier terrain with a
maximum displacement threshold of 50 cm d$^{-1}$. In addition, the mean surface velocity of the glacier
tongue (MSVT) was generated by averaging the velocities with the same threshold of pixels over the
glacier tongue region with an altitude of less than 5650 m where ice fall occurred for Longbasaba Glacier.
Unfortunately, there was no perfect Landsat pair to extract the glacier surface velocity for 2012, so the
surface velocity from 2011–2013 was calculated instead using the individual velocity maps for
2011–2012 and 2012–2013. Finally, the extra-annual characteristics of the MSVG and MSVT were
detected and discussed.
The monthly mean velocities indicate that the intra-annual variations in ice flow and were analyzed
from the GoLIVE (Global Land Ice Velocity Extraction from Landsat 8, Version 1) data set with a time
difference of 16 days during 2013–2019. The GoLIVE data set contains the glacier surface velocities
with a spatial resolution of 300 m and is available for free from the National Snow and Ice Data Center
(NSIDC) (Scambos et al., 2019). This data set was extracted using COSI-Corr and Landsat 8
panchromatic images obtained from 2013 to present (Fahnestock et al., 2015), and provides the glacier
surface velocities with a time interval of multiples of 16, for example, 16, 32, 48, and 64 days, with
accuracies ranging between ~1 m d$^{-1}$ to 0.02 m d$^{-1}$. In this study, we used a total of 55 ice velocity tiles
during the period of 2013-10-20 to 2019-1-6 to analyze the intra-annual characteristic of the surface
velocities of Longbasaba Glacier. However, the surface velocity values from June to September were not
enough to exhibit the ice flow pattern. Then, a quadratic polynomial fitting was performed to assess the
ice flow during specific months, and the horizontal movements of the glacier during different seasons
were determined.
**3.3 Basin morphology and water volume of the glacial lake**
*In-situ* measurements of the water depths of Longbasaba Lake were taken in September 2009 using a
comprehensive measuring system containing an echo sounder and a GPS receiver (Yao et al., 2012). A
total of 39, 558 echo sounder points with positions were collected. In addition, another 33 random points
were obtained using a measuring rope to evaluate the accuracy, which indicated an error of less than 2 m
for the water depth measurements.
Combining the water depth measurements and the lake boundary in 2009, the depth of each pixel
within the lake basin was obtained using the ordinary Kriging interpolation Method. In the Himalayas,
the majority of lake expansions occur with ignorable fluctuations of lake water level (Song et al., 2017;
Zhang et al., 2017), and thus, we assume that the water level of Longbasaba Lake remained stable during
the investigation period at the constant water level of 5499 m, which was measured by Yao et al. (2012).
Subsequently, the basin morphology of Longbasaba Lake in 2009 was reconstructed.
Since Longbasaba Glacier is a typical, huge valley glacier with a flat tongue, we assume that the
hypsography of the lake basin could approximately indicate the topography of the glacier bed close to
the terminal. Previous studies have revealed that large glacial lakes only form in areas where the glacier
surface gradient is less than 2º (Reynolds, 2000; Quincey et al., 2007). Thus, in this study, the contour
lines of the lake basin (interval of 20 m) were extracted and were subsequently extrapolated to the lake
basin with the maximum area in 2018 and a gradient of 2º. Finally, the basin morphology of the proglacial
lake in 2018, which has a spatial resolution of 30 m, was reconstructed based on these contour lines,
water depth measurements, and the lake boundary. Based on the basin morphology and lake boundaries
for the different years, the lake water volumes in each year, $V_l$, were estimated using the following
equation:
$$V_l = \sum_{i=1}^{N_l} h_l^i \, s, \tag{1}$$
where $N_l$ is the pixel number within the lake polygon in each year, $h_i^l$ is the depth of the individual
pixel, and $s$ is the area of an individual pixel with a value of 900 m$^2$.
**3.4 Mass contributions of glacier wastage to lake water volume**

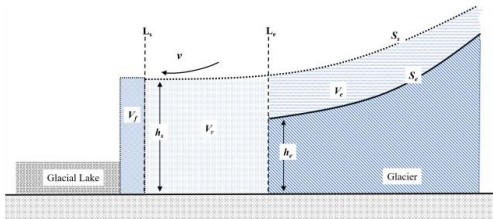

**Figure 2.** Mass budgets contributed by the glacier wastage to the lake water volume. $L_s$ is the glacier front in the
first year, and $L_e$ is the glacier front after retreat in the second year. $V_e$ is the glacier volume loss contributed by the
glacier surface lowing, $V_f$ and $V_r$ are the ice losses contributed by the ice flow and terminal retreat, respectively.
Due to the effects of climate warming and the formation of proglacial lakes, the mass wastage of lake-
terminated glaciers predominantly results from ice melt and avalanches and is characterized by terminal
retreat and surface lowering (Fig. 2). Consequently, the impacts of glacier change on lake water volume
can be divided into two portions: mass contribution from terminal motions, $M_t$, and mass contribution
that is not from terminal motions, $M_{nt}$. The former can be further divided into two phases: terminal
advance due to the ice flow of the glacier tongue and synchronous terminal retreat due to ice melt and
avalanches from the glacier front. Eventually, the impact of the glacier wastage on the lake water volume
can be classified into three synchronous components of mass budgets contributed by: (i) ice flow, $V_f$; (ii)
terminal retreat, $V_r$; and (iii) changes in the surface elevation over the area higher than $L_e$, $V_e$ (Fig. 2).
Then, the mass contributions of the glacier changes to the lake water volume were calculated using the
following equation:
$$M_G = M_t + M_{nt} = (V_f + V_r)\rho_{ice} + V_e\rho_{gla}, \tag{2}$$
where $\rho_{ice}$ is the density of the ice in the glacier tongue, and a constant value of 900 kg m$^{-3}$ was used
for the ice-to-mass conversation, as recommended by Kääb et al. (2012). $\rho_{gla}$ is the average density of
the glacier for a long-time scale estimation, with a value of $850 \pm 60$ kg m$^{-3}$ (Huss, 2013).





The ice volume contributed by the ice flow of the glacier terminal could be estimated using the
following equation:
$V_f = \frac{vt}{R} \sum_{i=1}^{N_f} h_s^i s,$     (3)
where $v$ is the average surface velocity of the glacier tongue; $t$ is the interval of the investigation periods;
$R$ is the spatial resolution of the pixel; $N_f$ is the pixel number within the profile of the glacier front; and
$h_s^i$ is the glacier thickness for an individual pixel in the previous year. The glacier bottom layer flows
slower than the upper layer, with a speed of about 30-80% of the surface velocity (Perutz, 1949; Mathews,
1959; Harper et al., 2001; Copland et al., 2003). In this study, we chose 70% of the MSVT as the value
of $v$. The thickness of the glacier terminal could be calculated by comparing the elevations of the surface
and the bed of the glacier front in a specific year. The surface elevation of Longbasaba Glacier in 1980
was extracted from the 1:50,000 Chinese historical topographic map (Wei et al., 2015; Wu et al., 2018).
By combining the position and bed elevation of the glacier front, the thicknesses of the glacier front from
1988 to 2018 were generated and modified using the average surface-lowering rate of the glacier tongue,
which were extracted from HMA_Glacier_dH data described in *Sect. 3.1* with the values of
approximately -0.89 ± 0.04 m a$^{-1}$ during 1975–2000 and -2.04 ± 0.04 m a$^{-1}$ during 2000–2016.
The mass volume contributed by the retreat of the glacier front was evaluated using the following
equation:
$V_r = \sum_{i=1}^{N_r} h_s^i s,$     (4)
where $N_r$ is the number of pixels over the terrain between the profiles of L$_s$ and L$_e$. The changes in the
ice volume contributed by lowering of the surface elevation, $V_e$, can be estimated using the following
equation:
$V_e = \overline{h_\Delta} s_g,$     (5)
where $s_g$ is the glacier area; and $\overline{h_\Delta}$ is the average lowering rate of the glacier surface higher than L$_e$.
The mean changes in glacier surface elevation during the two periods of 1975–2000 and 2000–2016
were extracted from the HMA_Glacier_dH data set.
**3.5 Accuracy analysis**
The geolocation errors of the pixels on the glacier/lake boundaries generated through a careful manual
approach that can be controlled with a subpixel accuracy of approximately 0.5 pixels. The accuracies of
the generated area are defined by the buffer around the glacier/lake perimeters and are equal to 0.5 pixels
multiplied by the pixel number within the perimeters and the spatial resolution of the images. According
to pan-sharpening employing principle-component analysis, Landsat TM\ETM+\OLI images can exhibit
an optimized usage with a spatial resolution to 15 m (Wu et al., 2018). Thus, the uncertainties in the
generated area of the glacier and lake are ~1% and ~28%, respectively. The accuracy of the main flowline
length was also controlled within 0.5 pixels (~± 8 m).
Based on the assumption that the outline of the glacier accumulation zone remained stable during the
investigation period, the areal measurements were compared to assess the image-image evolution of the
glacier and lake. The errors in the areal changes in the glacier and lake were relatively small for a control



approach and can be evaluated using the flowing equation (Krumwiede et al., 2014; Haritashya et al.,
2018):

$e_{ac} = n * R^2 / \sqrt{m},$  (6)
where $n$ and $m$ are the numbers of pixels and vertices, respectively, of the digital polygon defining the
change in the area during the specific period. Finally, the accuracy of the changes in the glacier and lake
areas was approximately ± 0.003 km².
The water volume was calculated by multiplying the lake depth and area, and then, the accuracy of the
estimated water volume was assessed using error propagation. The interpolated accuracy of the lake basin
depth is ± 4.87 m, which was obtained by comparing the interpolated points and the *in-situ* measurements.
Considering the accuracy in the lake depth was less than 2 m based on *in-situ* measurements, the final
accuracy of the lake basin depth is ± 5.26 m.
According to the error propagation, the accuracies of the mass contributions from the glacier terminal
motions and surface lowering were controlled by the errors in the glacier surface velocities, thicknesses,
and thinning rates. The estimation accuracy of the surface velocity was determined from non-glacier and
stable terrain in the investigated region, in order to eliminate the influence of bedrock movements. The
uncertainty in the glacier thickness was determined by the elevations of the glacier surface and the lake
basin. Both the accuracies of the topographic maps and the elevation lowering rate of the glacier tongue
determined the precision of the elevations of the glacier surface. The error in the lake depth was used to
assess the accuracy of the lake basin. The uncertainty in the glacier thinning rate and the elevation
lowering rate of the glacier tongue, depend on the precision of the HMA_Glacier_dH data set, which is
approximately equal to ± 0.04 m a⁻¹ for the two periods of 1975–2000 and 2000–2016.
**4 Results**
**4.1 Glacier retreat and lake expansion**
Longbasaba Glacier has experienced continuous and accelerating areal recession during 1988–2018, with
an inhomogeneous tendency toward terminal retreat in the different phases (Fig. 3). Overall, the glacier
area has decreased by 0.988 ± 0.093 km² since 1988, and had decreased to 29.551 ± 0.308 km² by 2018
with a mean decrease ratio of 3.23% (0.11% a⁻¹) during the past 30 years. Due to the parent glacier
degradation in area, Longbasaba Lake has expanded from 0.604 ± 0.209 km² in 1988 to 1.591 ± 0.389
km² in 2018, with an increasing ratio of 164% relative to the lake area in 1988. Before 2008, the glacier
area decreased with a dramatic fluctuation (Fig. 3). The greatest area loss occurred from 1993 to 1994,
causing an area of 0.089 ± 0.003 km² (0.29%) to disappear based on the total glacier area in 1993. During
the periods of 2000–2001 and 2003–2004, this glacier also experienced the most significant recession,
with areal changes of 0.078 ± 0.003 km² and 0.076 ± 0.003 km², respectively. However, the total area of
Longbasaba Glacier decreased by less than 0.05 km² during the other periods before 2008. In particular,
during the periods of 1988–1989, 1989–1990, 1991–1992, 1994–1995, and 2004–2005, this glacier
remained nearly stable with an area loss of less than 0.01 km². Longbasaba Glacier has retreated with a
relatively stable ratio during the recent decade, and the range of areal recession has varied from 0.026 ±





0.003 to 0.044 ± 0.003 km². Overall, the glacier has experienced accelerating shrinkage with a mean
areal recession rate of 0.032 ± 0.003 km² a⁻¹. Nevertheless, a decelerating tendency of glacier
degeneration in the area occurred during the most recent decade, but with a mean area loss rate of 0.035
± 0.003 km² a⁻¹, which is greater than the overall mean recession rate from 1988 to 2018.

The length of the main flowline of Longbasaba Glacier was 8274 ± 8 m in 2018, and decreased by

1577 ± 11 m (52.6 ± 0.4 m a⁻¹) from 1988 to 2018, with a mean recession ratio of 16.01% (0.53% a⁻¹)
relative to its length in 1988. The decreasing trend in length is similar to that of the glacier area (Fig. 3),
that is, the fluctuation in the changes during the last decade was significantly smoother than that during
1988–2008. The most dramatic length recessions occurred during the periods of 1993–1994, 2000–2001,
and 2003–2004 when the glacier experienced a length retreat of >180 m a⁻¹. Nevertheless, the main
flowline showed a slight change in the length within a pixel or remained nearly stable in the other periods
before 2008. From 2008–2018, the differences between the length changes were less than 22 m (55 ±
11–76 ± 11 m), except for the periods of 2014–2015 (32 ± 11 m) and 2017–2018 (42 ± 11 m). Overall,
the glacier experienced an increase in length recession during 1988–2018, similar to the trend in areal
recession, but with a higher mean length retreat (58.1 ± 1.1 m a⁻¹) than the period of 1988–2008 (49.8 ±
0.6 m a⁻¹). However, a slight decrease in the length recession rate was observed during the last decade.

The geodetic estimation from the HMA_Glacier_dH data set reveals continuous, decelerating mass

wastage for Longbasaba Glacier. Overall, the glacier surface elevation has decreased by -0.34 ± 0.04 m
a⁻¹ from 1975 to 2016. The lowering rate of the glacier surface was -0.38 ± 0.04 m a⁻¹ from 1975 to 2000,
contributing a total mass loss of 0.128 ± 0.014 km³ from 1988 to 2000. Based on the variations in the
glacier area, the thinning rate slightly decreased to -0.28 ± 0.04 m a⁻¹ from 2000 to 2016, releasing an
approximate mass budget of 0.138 ± 0.020 km³ during 2000–2018. Finally, Longbasaba Glacier has
exhibited an average mass balance of -0.27 ± 0.04 m w.e. a⁻¹ during the investigation period. In contrast,
the glacier tongue has experienced an accelerating lowering in surface elevation from 1975–2016, with
higher thinning rates ranging from -0.89 ± 0.04 m a⁻¹ before 2000 to -2.04 ± 0.04 m a⁻¹ from 2000 to

2016.


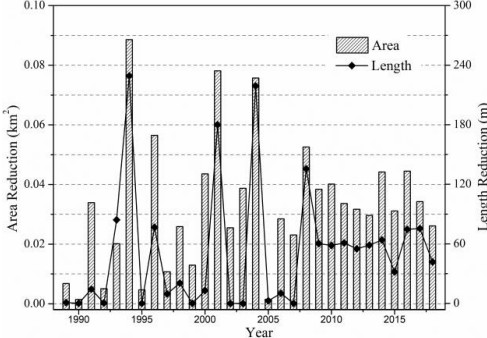

**Figure 3.** Comparison between changes in the area and length of Longbasaba Glacier during 1989–2018. The
changes in the glacier length was detected by combining the main flowline and front potions of the glacier. The
changes in the glacier area was estimated just by considering the motions of the glacier terminal.





It should be noted that the flanks of the glacier terminal retreated at a different rate than the center
before 2008, while the rates of the flanks and the center of the glacier were similar from 2008 to 2018
(Fig. 4). This is manifested by the fact that the amplitude and phase of the fluctuations in the changes in
the glacier area and length were not completely synchronous, and this mismatch indicates the specific
patterns of terminal retreat in different periods. For example, the fact that the glacier area decreased
significantly while the main flowline remained nearly stable means that the center of the glacier terminal
has experienced a slight retreat, while a huge area loss occurred on the flanks, which means that large-
scale ice avalanches at the flanks of the terminal were the main characteristics of the terminal retreat
during this period. By comparing the retreat amplitudes of the changes in the glacier area and length, the
patterns of terminal retreat were divided into three categories (Fig. 5):

*1) The area and length simultaneously and significantly decreased, for example, in 1993–1994,*

*2000–2001, and 2003–2004, huge ice avalanches occurred at the center of the glacier terminal*

*2) The area retreated significantly with nearly stable length, for example, in 1990–1991,1999–2000,*

*20001–2003, and 2005–2007, huge ice avalanches occurred at the flanks of the glacier terminal*

*3) The area and length decreased with similar fluctuation, for example, in 2009–2018, the glacier*

*terminal retreated as a whole due to small-scale ice avalanches*

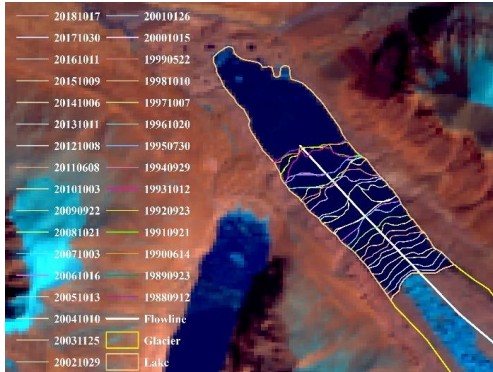


**Figure 4.** Variations in the front positions of Longbasaba Glacier generated from Landsat images during 1988–2018.
The background map is the Landsat OLI image taken in 17 October 2018. The white line shows the main flowline
of Longbasaba Glacier and exhibits the characteristic of changes in glacier length combined with the positions of
the glacier front.



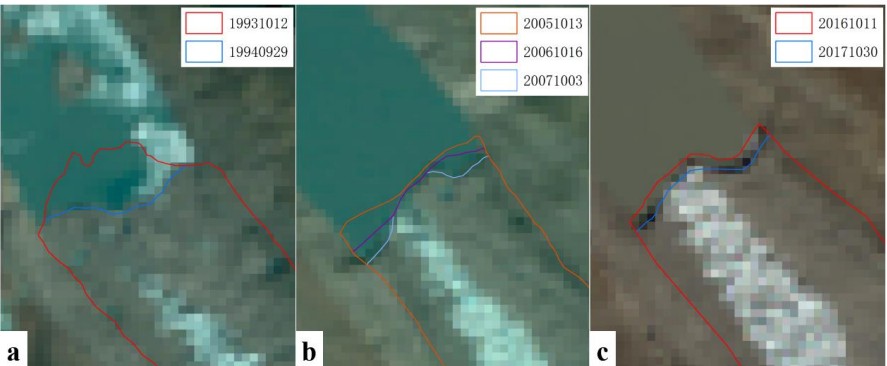

**Figure 5.** Three categories representing different patterns of terminal retreat of Longbasaba Glacier in specific periods. The background maps are Landsat TM\OLI images taken in 1994, 2007, and 2017, respectively. **(a)** shows the *Category 1*, with huge ice avalanches occurred at the center of the glacier terminal. **(b)** shows the *Category 2*, with huge ice avalanches occurred at the flanks of the glacier terminal. **(c)** shows the *Category 3*, with a whole terminal retreat for the glacier terminal.

These three categories of patterns of terminal retreat reveal different processes of ice masses entering the proglacial lake from the glacier terminal. *Categories 1* and *2* suddenly release numerous ice masses accompanied by debris cover into the glacial lake and potentially cause huge waves, which put pressure on the moraine dams and increase the failure risk of the moraine-dammed lake. In contrast, *category 3* releases small-scale ice avalanches with an insignificant mass budget from the glacier terminal, which would not obviously increase the risk of GLOFs for Longbasaba Lake.

### 4.2 Characteristics of the glacier surface velocity

The MSVG shows a decreasing trend during 1989–2018, with a similar trend for both the glacier and the glacier tongue (Fig. 6). The MSVT was $4.95 \pm 1.03$ cm d$^{-1}$ during the investigation period, which is significantly greater than the MSVG ($3.55 \pm 1.03$ cm d$^{-1}$), but it decreased more significantly later.

The fluctuation in the variations in the MSVT during the different periods was significant and was much greater than that of the MSVG, but both experienced synchronous fluctuations. The MSVTs were higher than the MSVGs during 1988–2018, except for 1989–1990 and 1997–1998, during which the MSVGs were slightly higher than the MSVTs. Consequently, a gentler fluctuation in the MSVG was found, according to the less significant fluctuation in the surface velocity of other zones compared to the glacier tongue. The trend in the changes in the MSVG does not agree with the changes in the glacier area and length, with correlation coefficients of less than 0.3. This reveals that the relationship between ice flow and the reductions in the glacier area and length is not clear.

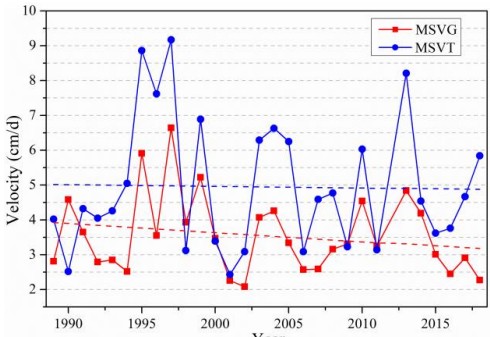

**Figure 6.** Mean surface velocities for the whole glacier (MSVG) and glacier tongue (MSVT) during the investigation

period of 1989–2018. The blue and red dotted lines were given by the linear fitting method and show the decelerating

ice flow for Longbasaba Glacier.

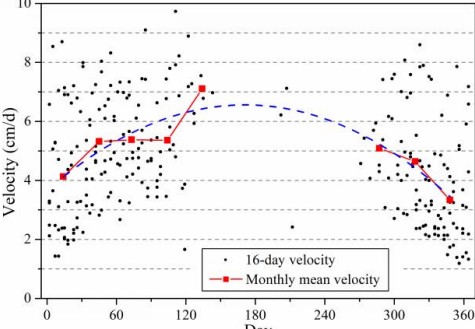

**Figure 7.** Intra-annual variations in the surface velocities of Longbasaba Glacier generated using GoLIVE. The black

points show the mean velocities in the middle of 16 days. The values of 16-day velocities in summer and autumn

were rare. The monthly mean velocities in summer and autumn were interpolated by applying the quadratic

polynomial fitting method (blue dotted curve) based on the monthly mean velocities in spring and winter.

The seasonal MSVGs were calculated using GoLIVE (Fig. 7). The MSVGs were distributed

predominantly in winter and spring (from October to May in the next year). Then, by applying a quadratic

polynomial fitting, we assessed the MSVGs in summer and autumn (from June to September). The fastest

ice flow occurred in summer (May, June, and July), and the velocity of the glacier surface decreased in

spring (February, March, and April) and autumn (August, September, and October), with a flow rate of

86% and 89% of the summer velocity, respectively. The slowest glacier surface movements occurred in

winter (November, December, and January in the next year) when the ice flowed at a ratio of just 62%

and 73% compared to the MSVGs in the summer and the annual average velocity, respectively.

**4.3 Changes in the water volume of the glacial lake**

Based on the estimated basin morphology of Longbasaba Lake (Fig. 8a), the maximum depth of the

glacial lake was $99.52 \pm 5.26$ m in 2018. A continuously increasing trend in the mean depth of the glacial





lake before 2010 was accompanied by a slight decrease during the last decade, due to an elevation rise
with a slight slope and the narrower width of the lake basin.

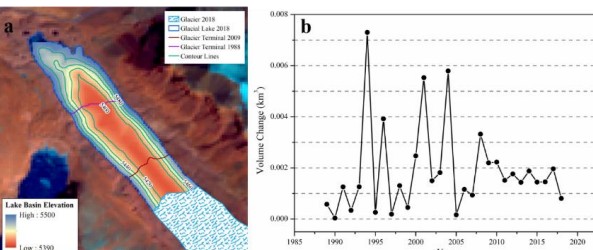


**Figure 8. (a)** Basin morphology of Longbasaba Lake reconstructed based on the echo sounder points in 2009 and
the lake boundary in 2018. The background map is the Landsat OLI image taken in 2018. The basin morphology
within the lake boundary in 2009 was generated directly from the *in-situ* measurement points and the other part was
extrapolated. The deepest point (5400 m) is located at the center of the glacier terminal in 1988. **(b)** Volume changes
of Longbasaba Lake estimated based on the basin morphology and changes in the lake boundary during 1988-2018.
Combing the lake depth and outlines, the water volume of the glacial lake approached a maximum
value of $0.080 \pm 0.022$ km$^3$ in 2018. The water volume of the proglacial lake has increased by 233%
$(0.002 \pm 0.001$ km$^3$ a$^{-1})$ from 1988 to 2018. The most significant expansions in the lake volume occurred
in the three periods of 1993–1994, 2000–2001, and 2003–2004, with expansion rates of $>0.005$ km$^3$ a$^{-1}$.
These periods agree with the time when the glacier front retreated as described by *Category 1*. In addition,
the studied lake experienced insignificant changes $(< 0.0005$ km$^3$ a$^{-1})$ in water volume during the periods
of 1989–1990, 1991–1992, 1994–1995, 1998–1999, and 2004–2005. According to the dramatic
fluctuation in the variations in the changes in the glacier area, the differences in the lake volume during
different periods were significant before 2008 (Fig. 8b). However, the increasing water volume slowed
slightly from 2008 to 2018.
**4.4 Mass contributions of glacier shrinkage to lake water volume**
The mass budget of glacier motions was predominantly contributed by the glacier surface lowering (Fig.
9). The change in the glacier surface elevation contributed more than 80% of the total mass contributions
from glacier reduction including ice melt and avalanches. In particular, from 1989 to 1990, more than
90% of the mass budget resulting from glacier shrinkage was contributed by elevation changes in the
glacier surface. During 1993–1994, 2000–2001, and 2003–2004, when Longbasaba Glacier retreated as
described in *Category 1*, the proportions of the mass contributions from the lowering of the glacier
surface decreased to approximate 50%, which suggests that the mass contributions from the glacier
motion have increased during these periods. According to the decrease in the glacier area and surface
lowering rate, the mass wastage from the changes in the glacier surface elevation continuously decreased
by 30%, from $0.0099 \pm 0.0011$ km$^3$ during 1988–1989 to $0.0070 \pm 0.0011$ km$^3$ during 2017–2018. As
the glacial lake expanded continuously, the ratio of the mass contribution from lowering of the glacier

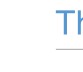
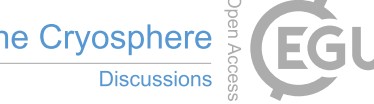

surface to the lake volume has decreased significantly, from 41% during 1988–1989 to 9% during
2017–2018, and it exhibits a decreasing trend with a greater slope before 2000 than in the last decade.
Mass contributions from ice flow from the glacier terminal were approximately 10% of those due to
lowering of the surface elevation, followed by an obvious fluctuation with an average mass budget of
$0.0008 \pm 0.0002$ km$^3$ during the investigation period. In particular, during 1994–1995, 1996–1997, and
2003–2004, the fastest ice flow resulted in the most mass wastage (more than 0.0015 km$^3$). Overall, a
slight decrease occurred in the mass contributions from ice flow during the last 30 years. The ratios of
the ice flow contribution to lake volume were less than 5% during 1989–2018, which was accompanied
by a slight decrease with significant fluctuations. In addition, the ratios during the last decade were less
than 1%, except during 2009–2010 and 2010–2012. These results suggest that ice flow from the glacier
terminal played a slight role in increasing the water volume of the proglacial lake.
Responding to huge fluctuations in areal changes in Longbasaba Glacier during the investigation
period, the mass budgets resulting from the retreat of the glacier front varied significantly, with an
average mass contribution of $0.0021 \pm 0.0003$ km$^3$, ranging from $0.0076 \pm 0.0005$ km$^3$ during 1993–1994
to nearly zero during 1989–1990 and 2004–2005. The terminal retreats according to the pattern of
*category 1,* contributing a significantly greater mass budget to the lake volume than the other patterns,
with mass contributions of > 0.0065 km$^3$ and ratios of >10% to the lake volume. In addition, the mass
contributions from the retreat of the glacier front as described in *categories 2* and *3* were
indistinguishable.
Overall, glacier shrinkage released an average mass of $0.0111 \pm 0.0016$ km$^3$ into the glacial lake with
a slightly decreasing trend. The most mass contributions occurred during 1993–1994, 2000–2001, and
2003–2004, when the glacier terminal retreated as described in *category 1*; while glacier shrinkage as
described in *category 2* contributed a larger mass budget relative to *Category 3*. During the last decade,
the ratios of glacier change to lake volume were less than 16% with a mass wastage of less than 0.010
km$^3$. These results indicate that ice avalanches from the glacier terminal were an important source for
lake expansion and were the predominant factor in the rapid mass wastage of the lake-terminated glacier.

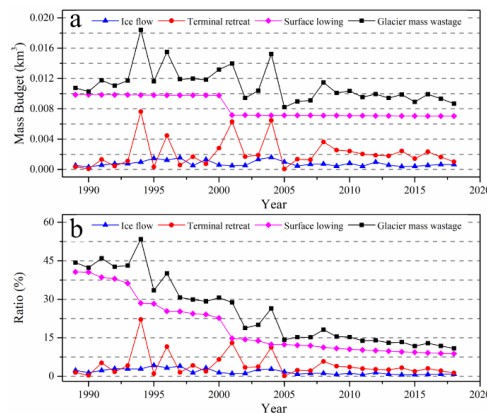





**Figure 9.** Mass contributions **(a)** and ratios **(b)** of different patterns of glacier shrinkage to glacial lake water volume during 1989–2018. The black lines show the total mass contributions and ratios of glacier shrinkage to lake water volume.

**5 Discussion**

**5.1 Uncertainty in the estimated mass contributions of glacier shrinkage to lake water volume**

The variations in lake water volume were estimated based on the assumption that the proglacial lake experienced a negligible interannual change in water level during the investigation period. This ideal assumption is consistent with that made in previous studies. From 2003 to 2009, an insignificant average increase rate of 0.21 m a$^{-1}$ was found in lake levels over the Third Pole based on ICESat altimetry data (Zhang et al., 2011; Zhang et al., 2017). The glacial lakes in the Himalayas exhibited no statistical variations in water level during the last decade based on the ICESat and CryoSat-2 data, although debris-contacted proglacial lakes have a slightly higher average increase rate of 0.8 m a$^{-1}$ (Song et al., 2017). For proglacial lakes with the stable moraine dam and outlet, their small fluctuations in water level are ascribed to natural outflow regulations (Song et al., 2017). Based on several *in-situ* measurements, the water level of Longbasaba Lake showed an insignificant fluctuation (Yao et al., 2012; Wang et al; 2018). According to the mean lake depth of ~50 m, a mean change of 0.8 m a$^{-1}$ in lake water level could contribute errors of ~1.5% relative to the estimated water volume in this study.

Mass wastage from lake-contacted glacier is composed of three components: evaporation/sublimation into the air, infiltration into the ground, and ice melt/avalanche into the proglacial lake. Mass budgets determined without considering the first two components would overestimate the mass contributions of the glacier changes to the lake water volume. Nevertheless, for glaciers developed in high mountain regions, the groundwater system and evaporation/sublimation in the glacier zone has a negligible impact on glacier hydrology and water resources relative to ice melt and avalanches (Kang et al., 1999; Brock et al., 2010; Liu et al., 2010; Zhang et al., 2012). Consequently, the methods used in this study, that is, ignoring evaporation/sublimation and infiltration, are reasonable and can be used to precisely reconstruct the mass contribution series of glacier degeneration to proglacial lake water volume.

Without considering the impacts of hypsometry and local topography, a mean lowering rate of -3.9 m a$^{-1}$ was found for the glacier front elevation in the Himalayas (Song et al., 2017). In this study, we determined the glacier front elevation by combing the front position and the surface lowering rate and determined a precise series of elevations for the glacier front. Nevertheless, the mean surface lowering rate of the glacier tongue was used to indicate the elevation changes in the glacier front where the surface elevation commonly decreased with a higher ratio than in other locations. The assessed thicknesses of the glacier fronts were slightly overestimated in this study, which subsequently resulted in an overestimation of the mass contribution of the terminal motions.



### 5.2 Mechanism of the variations in lake-terminated glacier shrinkage


The rapid mass wastage of glaciers causes rapid expansions of the glacier-contacted lakes (Fujita and
Sakai, 2014; Immerzeel et al., 2014), which could expedite ice mass loss for parent glaciers (Gardelle et
al., 2013; King et al., 2017). This is supported by the fact that proglacial lakes in the Himalayan range
have experienced a rapid expansion extent of 36.5% during 2000–2014, which is more dramatic than
that of other glacial lakes (Song et al., 2017). The lake-terminated glacier also showed an accelerating
and greater mass melt than other glaciers, with mean glacier thickness changes of -0.65 ± 0.04 m a$^{-1}$
during 1974–2000 and -0.80 ± 0.05 m a$^{-1}$ during 1974–2000 in the Poiqu River Basin (Zhang et al.,
2019). Nevertheless, Longbasaba Glacier has experienced an accelerating area decrease but accompanied
by a decelerating and moderate glacier surface lowing.
Glaciers in the Himalayas are more sensitive to climate change than glaciers in other mountain ranges
with higher annual temperatures, for example, Karakoram. In particular, a more significant response was
found in the central Himalayas than in the western Himalayas and Karakoram (Fujita, 2008; Sakai et al.,
2015). Most glaciers in the central Himalayas received their maximum accumulation in summer because
of high monsoonal precipitation and high elevations (Ageta and Higuchi, 1984; Yao et al., 2012; Azam
et al., 2018). Temperatures in the central Himalayas have increased significantly since 1960. A warming
rate of 0.024 ± 0.004ºC a$^{-1}$ was observed at Nyalam station during 1967–2017, followed by a decrease
in precipitation of -0.76 ± 1.34 mm a$^{-1}$ during 1960–2013 with a heterogeneous pattern (Zhang et al.,
2019). Throughout the Himalayas, the observed glacier wastage is consistent with increasing temperature
and decreasing precipitation (Azam et al., 2018). Along the Himalayan range glacier areal recession was
more moderate than overall throughout High Mountain Asia over the last five to six decades, with a high
variability in rates ranging from -0.07% a$^{-1}$ to -1.38% a$^{-1}$ and a mean retreat rate of -0.36% a$^{-1}$ for
1960–2010 (Cogley, 2016; Azam et al., 2018). The average mass balance estimated using the geodetic
method was less negative over the Himalayas than the global mean (Kääb et al., 2012; Gardelle et al.,
2013), exhibiting a mass loss of -0.37 m w.e. a$^{-1}$ between 1962–2015 (Azam et al., 2018). The central
Himalayas have more gentle mass wastage than other regions in the Himalayas (Gardelle et al., 2013;
King et al., 2017). In the Poiqu River Basin in the central Himalayas, glacier area has decreased by -0.52
± 0.05% a$^{-1}$ during 1964–2000 and increased to -0.72 ± 0.08% a$^{-1}$ after 2000 (Zhang et al., 2019). For
glacier surface elevation, an overall decrease of -0.38 ± 0.18 m a$^{-1}$ occurred during 1974–2000,
accompanying a more negative rate of -0.40 ± 0.14 m a$^{-1}$ during 2000–2017 (Zhang et al., 2019).
Nevertheless, compared to other lake-terminated glaciers in the central Himalayas, Longbasaba Glacier
showed a specific response to climate change under the same pattern of climate change conditions, which
suggests that other factors besides climate change play an important role in glacier recession.
Debris cover affects glacier mass budgets by controlling the heat conduction mechanism over the
glacier surface, which depends on its thickness and the nature of the debris cover (Potter et al., 1998;
Konrad et al., 1999; Brock et al., 2010; Reid and Brock, 2010; Lambrecht et al., 2011; Nicholson and
Benn, 2013). However, several previous studies have determined that the surface-lowering rates of
debris-free glaciers and debris-covered glaciers were accompanied by similar amounts of mass wastage
in response to climate change (Kääb et al., 2012; Nuimura et al., 2012). In addition, the mass reduction




of debris-covered glaciers is predominantly manifested as surface lowering without significant frontal
retreat (Rowan et al., 2015; Banerjee, 2013). The internal ablation over the debris-covered tongue, for
example, enlargement of englacial conduits, has a direct and/or indirect effect on the mass budgets
through ice melt and collapse on the surface (Thompson et al., 2016; Benn et al., 2017). For lake-
terminated glaciers with debris-covered tongues, fine-grained thick and intact debris cover insulates the
ice from solar radiation, but this effect could be counteracted by significantly enhanced interaction
between the glacier and lake in thermokarst features (Sakai et al., 2002; Buri et al., 2016; Miles et al.,
2016; Watson et al., 2016). Under the same climate change conditions, Longbasaba Glacier showed a
decreasing mass wastage during recent decades, which was followed by a contrary trend in the glacier
tongue. Consequently, the debris cover of the Longbasaba Glacier played an important role in the rapid
recession of the glacier tongue, but a further dynamic study is needed to assess the effect of the process.
Glacier surface velocities fluctuate with mass budgets at the decadal scale (Span and Kuhn, 2013;
Dehecq et al., 2018). The ice flow of the glaciers in High Mountain Asia has commonly decreased, and
a dramatic decreasing amplitude occurred in the Himalayas (Dehecq et al., 2018). In the Poiqu River
Basin, the majority of lake-terminated glaciers exhibited faster ice flow than the other glaciers, which
was followed by heterogeneous variations controlled by the topographic features of the glacier terminal
(Zhang et al., 2019). This overall decrease in ice flow during a period of rapid glacier shrinkage suggests
that mass loss commonly prompts glaciers to adjust their dynamics (Azam et al., 2012; Rowan et al.,
2015; Bhattacharya et al., 2016). Recently, several glaciers in Karakoram have shown accelerating ice
flow caused by positive mass budgets (Quincy et al., 2009; Azam et al., 2018). Unfortunately, the direct
relationship between the changes in the surface velocity and the mass balance is not evident at the glacial
or regional scales, which could be due to a lag in the response of ice flow to climate change (Heid and
Kääb, 2012; Vincent and Moreau, 2016; Dechecq et al., 2018; Vincent and Moreau, 2016). In addition,
the fluctuations in the velocity changes of Longbasaba Glacier displayed no obvious relationship with
the changes in glacier area, length, and mass balances.
Longbasaba Glacier, which is a typical lake-terminated glacier with a debris-covered tongue, has
experienced continuous and accelerating decrease in glacier area, but a decelerating mass wastage during
the past three decades, which does not agree with the overall trends in the changes in ice mass for glaciers
in the Himalayas and throughout the High Mountain Asia (Azam et al., 2018; Kääb et al., 2015;
Bajracharya et al., 2015). In addition, the glacier tongue exhibited a contrary trend in mass loss relative
to the glacier during the investigation period. For glaciers in direct contact with proglacial lakes, specific
changes in area and ice mass are controlled by a complicated combination of processes and diverse local
topographic conditions (Fujita and Sakai, 2014; Immerzeel et al., 2014; King et al., 2017; Song et al.,
2017; Zhang et al., 2019). To assess the complex processes of the changes in the area and ice mass of
lake-contacted glaciers, further detailed dynamic studies at the glacier-scale of the mass/energy
interaction between glaciers and glacial lakes are urgently needed in the future.



### 5.3 Potential triggers of GLOF for Longbasaba Lake

In the Himalayan range, GLOF events are predominantly triggered by the failure of moraine dams, caused by overtopping and/or self-destruction (Chen et al., 2006; Westoby et al., 2014; Rounce et al., 2017), which can potentially cause devastating disasters by transporting large amounts of debris (Allen et al., 2015). Large mass movements suddenly entering proglacial lakes, for example, ice/snow avalanches (Xu, 1987; Awal et al., 2010), rock falls and rockslides (Richardson and Reynolds, 2000), and extreme heavy precipitation (Harrison et al., 2018), can cause huge waves and the subsequent overtopping of bedrock or ice-core dams. The self-destruction of moraine dams is induced by piping/seepage, degradation of the ice-cores in the dams, dam collapse, and other triggers, for example, seismic events (Richardson and Reynolds, 2000; Chen et al., 2007; Westoby et al., 2014; Rounce et al., 2017; Harrison et al., 2018; Nie et al., 2018).

Current climate warming plays a predominant role in the degeneration of permafrost and ice cores in moraine dams, which can be a trigger for the failure of moraine dams (Vilímek et al., 2014; Harrison et al., 2018). In addition, extreme heat and extreme rainfall are potential triggers of GLOF events. Although no evident relationship was observed between climate change and an increase in GLOF events, it is predicted that the frequency of GLOF events increases during the next few decades by considering the lag times in the expansion and evolution of proglacial lakes (Harrison et al., 2018). Furthermore, under specific climate changes and their influence on hazards, complex glacier-proglacial lake interactions make glacier hazard study a challenging approach, but it is urgently needed (Marzeion et al., 2014; Shugar et al., 2017).

Ice avalanches with large masses from steep glacier fronts have been the predominant trigger of moraine dam failure in the Himalayan range (Awal et al., 2010; Nie et al., 2018) and have increased the potential for GLOFs (Sakai et al., 1998; Wang et al., 2008; Sakai et al., 2009; Gardelle et al., 2011). For Longbasaba Lake, numerous residual, various sized ice floes on the surface and bank of the lake were commonly observed by both *in-situ* measurements and remote detection during the investigation period and reflect the fact that ice avalanches commonly occurred (Yao et al., 2012; Wang et al., 2018). The glacier front retreat as described in *category 1*, for example, 1993–1994, 2000–20001, and 2003–2004, raised the lake water level by 6–11 m, assuming that the terminal motions released all of the mass of the ice avalanche at once and ignoring the influence of debris. While *category 2* retreat (e.g., 1999–2000, and 2001–2003) causes a water-level rise of more than 3 m. The recession as described in *category 3* can potentially raise the lake level by less than 2 m, especially after 2008. Ice avalanches create large amounts of ice and companying debris masses, which suddenly enter the proglacial lake and potentially create impact waves that trigger overtopping and failure of moraine dams. Although contributing much more masses than the glacier terminal motions, the glacier surface lowing released ice masses gradually and provided a slight increase in lake water level due to the stable outlet downstream of the lake basin. This suggests that ice avalanches potentially play a predominant role in the outburst risk of Longbasaba Lake. Piping/seepage in dams, rock-falls, and rockslides were observed infrequently (see Fig. 1c in Wang et al., 2018). However, these processes were considered to be negligible relative to ice avalanches from the glacier tongue.





Previous studies have revealed that the frequency and impact of GLOF events declined during the
most recent decades in the Himalayan range (Harrison et al., 2018). This is partly ascribed to several
successful measures carried out by local governments and communities to decrease the outburst
possibility of glacial lakes, including stabilizing moraine dams and fluvial systems (Carrivick and Tweed,
2013). The moraine dams and outlet of Longbasaba Lake are monitored and maintained by local
governments and scientists every year (Wang et al., 2018), including widening the river channel and
stabilizing moraine dams and banks of the river channel before 2009. In 2010, during a small-scale ice
avalanche, the moraine dams at the outlet of Longbasaba Lake were partly damaged, but did not fail.
Eventually, the GLOF possibility of this proglacial lake has declined even during a time of rapid recession
of the parent glacier, which may be manifested by the lagging response of Longbasaba Lake to climate
change over a long-time scale (Harrison et al., 2018).
**6 Conclusions**
The evolution records for the shrinkage of Longbasaba Glacier and the expansion of the proglacial lake
extend from 1988 to 2018, and the mass contributions from glacier shrinkage to the lake water volume
were assessed and analyzed.
During the past three decades, Longbasaba Glacier has experienced a continuous and accelerating
recession in the glacier area accompanied by the decelerating thinning and ice flow over the glacier
surface. Consequently, the extent and water volume of Longbasaba Lake had expanded significantly at
an accelerating rate. Lowering of the glacier surface played a predominant role in the mass contribution
from glacier shrinkage to the lake water volume and was an order of magnitude higher than those from
the motions of the glacier tongue. Overall, the mass contribution slightly decreased during the
investigation period with dramatic fluctuations before 2008 due to a combination of a decelerating
lowering rate of the glacier surface elevation and an accelerating decrease in the glacier area.
Longbasaba Glacier retreated with frequent ice avalanches, which suddenly released large amounts of
ice into the proglacial lake and became the main potential trigger for failure of moraine dams and
subsequent GLOF events. According to the areal expansion, decreasing mass contributions from the
parent glacier shrinkage, and several improvements to the drainage system by local governments, the
potential risk of GLOFs at Longbasaba Lake has continuously decreased during the last decade.
*Code and data availability.* The Landsat TM\ETM+\OLI images are available from the United States
Geological Survey (USGS). The High Mountain Asia Gridded Glacier Thickness Changes from Multi-
sensor DEMs, Version 1 is available from the National Snow and Ice Data Center (NSIDC) (Maurer et
al., 2018). The free software module Co-registration of Optically Sensed Images and Correlation (COSI-
Corr) is available from the Caltech Tectonics Observatory (TO) (Leprince et al., 2007). The Global Land
Ice Velocity Extraction from Landsat 8 (GoLIVE), Version 1 is available from the NSIDC (Scambos et
al., 2019). The *in-situ* echo sounder points with positions and water depths of Longbasaba Lake will be
provided by Junfeng Wei upon request.



*Author contributions*. JW and SL designed the study; JW, TZ, XW and YZ provided the formal analysis
and validations. All authors discussed the results and contributed to the writing and editing of the
manuscript.
*Competing interests*. The authors declare that they have no conflict of interest.
*Acknowledgements*. This work was founded by the fundamental programme of the National Natural
Science Foundation of China (grant no. 41701061, 41761144075, and 41771075), the research project
of Hunan University of Science and Technology (grant no. E51669), and the National Key Research and
Development Program of China (grant no. 2018YFE0100100). We are grateful to the California Institute
of Technology (CIT) for providing the not-for-profit institution with the COSI-Corr software.

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
