# Peer review of "Terminal motions of Longbasaba Glacier and their mass 1 contributions to proglacial lake volume during 1988-2"

_The Cryosphere, 2019_

## Referee Comment (RC1) · Anonymous Referee #1 · 20 Feb 2020

Junfeng Wei et al. studied the interaction of a proglacial lake and its parent glacier (Longbasaba Glacier, Chinese Himalaya) from 1988 to 2018. Is a matter of ongoing debate among glaciologists whether and how much glacier lakes contribute to glacier mass loss, and vice versa, how much glacier melt adds to glacier lake growth. The manuscript thus addresses a relevant question within the scope of TC. Junfeng Wei et al. use recently published, freely available data sets, most of them being derived from the Landsat satellite, to monitor changes in lake area, glacier area, glacier flow velocity, and glacier elevation. They aim at quantifying the individual shares in the total glacier mass loss to estimate their contribution to the well-documented increase in lake volume. Though I highly acknowledge the authors' effort to split the mass loss

at Longbasaba Glacier into different components, some doubts remain whether this goal could be met with the methodological framework presented. First of all, as far as I understood, the authors mostly use gridded data, for example on glacier flow velocities. Yet, they mostly use the mean of the entire distribution, instead of integrating over all observed values, so that they lose substantial parts of information. For glacier surface lowering, they seem to use only two values (pre and post 2000) from published data, which they assume to remain constant over the study period. Though the authors strive to provide the uncertainties for each dataset in remarkable detail, it remains difficult to infer how these uncertainties are propagated, when the authors calculate the individual contributions to glacier mass loss. Some of the lake-related factors such as water temperature, wind direction, or rising lake levels, (possibly promoting undercutting of the glacier front) are deemed negligible without formal proof. Additional key inputs such as snowmelt and rainfall, and outputs such as evaporation and lake outflow, are disregarded, so that the overall goal of quantifying the 'mass contributions of glacier wastage to lake water volume' seems oversimplified.

A considerable body of previous research has monitored in detail glacier retreat and corresponding lake growth both on local and basin-scale (Haritashya et al., 2018; Sharma et al., 2018; Somos-Valenzuela et al., 2014; Wang et al., 2015a, 2018; Watanabe et al., 2009; Xin et al., 2008). In this regard, the authors could use the introduction to clearly stress the novelty of their study by revealing questions and research gaps that remain from previous work. In general, I feel that the introduction could prepare the reader better for the following chapters. At the current stage, the introduction misses a stringent regional overview on glacier mass balances, lake dynamics, and methods to track these changes. For example, why is it necessary to monitor glacier flow velocities, if we want to quantify lake volumes? Which of the components in glacier mass loss have only rarely been addressed in previous studies? Some of this information is provided in the discussion, which could better serve in the introduction. In the discussion, the authors review some important mechanisms that occur at the interface between proglacial lakes and their parent glaciers. Yet this information is largely detached from their own findings, and they mostly point towards future research instead of showing possible solutions. For example, Tsutaki et al. (2019) recently modelled the mass balance of debris-covered glaciers in conjunction with an ice dynamical model to explain the thinning pattern of lake- and lake terminating glaciers. How can such physically based models help to understand the detailed observations at Longbasaba Glacier?

A final remark pertains to the issue that the authors mostly use qualitative descriptions of their own and others' findings. For example, the abstract has remained vague with regards how much Longbasaba Glacier retreated in volume, length, and area, and how much the lake volume increased in the same period. Yet these are key data that a reader expects to learn while reading those lines. The remainder of the manuscript contains 4, 10, and 14 times the terms 'insignificant', 'significant', and 'significantly', respectively, without a test of significance and mostly without figures or numbers underlining the significance of these findings. A similar critique applies to the repeated use of 'small' 'large', 'huge', 'typical', 'dramatic', 'slight(ly)', 'unfortunately', and 'different', with little relation to measured quantities in this or previous studies. I strongly encourage the authors to put more emphasis on objective language. I commented below on a sample of phrases where this is most obvious, though subjective terms should be replaced throughout the manuscript. Finally, I would be happy if the authors thoroughly revise their manuscript for typos and incorrect citations. In the following, I provide more detailed comments and questions to the points raised above.

Major comments

L14-16: A wealth of studies has claimed that there could be a link between lake expansion and outburst floods (Nie, 2017; Nie et al., 2018; Richardson and Reynolds, 2000; Rounce et al., 2016). Can the authors list some of such studies, which support this notion? Yet, to my best knowledge I cannot remember that an outburst flood in the Himalaya has been successfully predicted from a rapidly expanding moraine-dammed lake. I would thus caution against crediting Longbasaba Lake, or any other growing

lake, per se 'a high outburst risk'. Please note that 'hazard' and 'risk' in the authors' and previous studies has been used interchangeably, though these terms cover different aspects. Hazard is the probability of an event occurring, whereas risk is the product of hazard, vulnerability and the elements at risk. Please make sure to use the correct terms at any given location in the manuscript. Most aspects here seem to address hazard, not risk.

L20-22: Please quantify 'continuous and accelerating recession in glacier area and length' and 'decelerating surface lowing and ice flow'.

L22: How do the authors determine 'significantly' here? How do they measure that lake growth has accelerated?

L23: How much did surface lowering contribute to glacier downwasting?

L27: Does the issue that 'the potential risk of outburst for Longbasaba Lake has continuously decreased during the last decade' not contradict the main motivation of the study?

L29-30: When did glaciers start to lose mass? And how much? Would it make sense to review here studies that focused on glacier mass loss in the Himalayas?

L31: 'Reduction' in which quantity?

L33: How did the authors and the cited studies measure, and demonstrate increases in, 'potential risk and destructiveness' of GLOFs?

L35: Please clarify what the 'mass/energy interactions of the thermal regime' are. I suggest to split L34-38 into two or three new sentences.

L38-39: What is 'more significant shrinkage' compared to? How do the authors measure a 'significant mass budget'? And again, how do they quantify 'increases [in] the risk of GLOFs'?

L40: Can the authors provide examples on these 'ruinous impacts' for each of the

sectors they mention?

L42: How do the authors define the 'Third Pole'? Is it different from 'High Mountain Asia' (Brun et al., 2017; Dehecq et al., 2019)?

L44 and L49-50: Similar to the notions on increasing GLOF risk, I would caution against using the term 'potentially dangerous glacial lakes'. There is no common definition to my best knowledge, which variables classify a lake as 'potentially dangerous', nor is there a proof of concept. Please see the introduction in Rounce et al. (2016) on confounding estimates on the number of 'potentially dangerous lakes' in the Himalayas.

L45-46: Veh et al. (2018) did not report 70 GLOFs, and Khanal et al. (2015) is missing in the references. Also, please make sure to mention that the review of previous work here exclusively focuses on moraine-dammed lakes. In essence, more GLOFs have been documented from ice-dammed lakes, mostly in the Karakoram, than from moraine-dammed lakes (Bhambri et al., 2019).

L50-51: What are the expected losses from a GLOF of a given magnitude from Longbasaba Lake? Which countries will be affected in case of an outburst?

L52-55: From the reasons given here, I struggle to see why monitoring of glacier and lake area could help to predict outburst floods. Btw: what do the authors mean by this 'prediction': The timing, the magnitude, both?

L55-58: Please reformulate the scope of this study in at least two sentences.

L64: Please elaborate what a 'serac cluster' is, what the authors mean by 'numerous', and where the reader can see seracs and supraglacial lakes in Fig. 1b.

L67: 'remains at a high risk of outburst': contrary to this statement the authors wrote in L27: 'the potential risk of outburst for Longbasaba Lake has continuously decreased during the last decade'. Please clarify.

L72 and L74: Are the values in m a-1 average or maximum rates? Additionally, how

do the authors define 'rapid areal expansion'? Is the expansion at Longbasaba faster than reported on regional or local scales (Nie, 2017; Song et al., 2016, 2017; Wang et al., 2015b)? Could above-average expansion (including increasing possible changes in flood volumes) be a suitable motivation for the authors' study? I would be happy to see these issues clearly emphasized in the introduction.

L111-L113: Why are the differences in accuracy so large between the two methods? Do the authors mean 'Brun' instead of 'Burn'? Which part(s) of their method did the authors apply to which dataset? Please elaborate more clearly.

L118-L122: From this description, I cannot follow what and how large the sliding window is; how signal and noise are defined and which measure of 'quality' is assessed here.

L124: What happens with data above (or below?) this maximum displacement threshold? Also, what additional information do the authors expect to gain from the surface velocity of the entire glacier, compared to the velocity of the tongue? Arguably, the velocity of the tongue could be a better indicator of what is happening at the terminus and the adjacent lake.

L129-130: Please elaborate what the authors mean with 'extra-annual characteristics' and what the authors 'detected and discussed' in this regard.

L135-138: Are the accuracies of the GoLIVE product global values or specific for Longbasaba glacier?

L141: Why do the authors use a quadratic fit instead of a linear?

L151: Please elaborate the choice of the parameters for Kriging.

L152: Quincey et al. (2007) wrote that "supraglacial lake formation is prevalent where glacier surface gradients are less than 2° from the glacier terminus". This statement is slightly different to the assumption here, given that a lake at Longbasaba Glacier already exists. Why should the growth of Longbasaba Lake stop at a surface gradient

of, say, 3°?

L159-161: 'the contour lines of the lake basin': are these the isobaths, i.e. the lake bathymetry? If so, why do the authors need to apply the 2° surface gradient threshold to clip the lake extent, if they already know the lake area from satellite images? Please reformulate L159-163 more clearly.

L196-197: It seems reasonable that the authors considered a different velocity for the entire ice column than for the values measured at the surface. But why do the authors choose exactly 70%? I guess the reported range from 30% to 80% relates to factors such as the gradient of the glacier bed, the glacier mass budget, or whether the glacier is cold or warm based. Please elaborate the choice of this value more clearly, and if necessary, please test how sensitive Vf is to this prefactor.

L198-199: It would be good to test whether previous estimates of glacier ice thickness (see Farinotti et al., 2019) match with the method from the authors. (Linsbauer et al., 2016) argue that ice thickness models allow for estimating the glacier bed topography, which could help modelling the evolution of Longbasaba Lake.

L223-224: 'Based on the assumption that the outline of the glacier accumulation zone remained stable during the investigation period': This is a strong assumption, given that the glacier has retreated continuously, so that the accumulation area has likely shrunk in the study period. Could mapping the snow line help? In any case, how does this part fit to the remainder of the sentence?

L225—226: What do the authors mean by 'The errors in the areal changes in the glacier and lake were relatively small for a control approach'?

L228-229: What is R? Why do the authors consider the number of vertices in the lake polygon? Is the number of vertices not a rather arbitrary value depending on how carefully the image interpreter digitizes the lake boundary?

L234: How did the authors obtain this measure of accuracy?
L252: What is 'a mean decrease ratio'? Do the authors mean the percentage loss of total glacier area per year?

L257: Did not the period 1993-1994 show 'the most significant recession'?

L259: A question that has been left untouched here and in the discussion is why the variance in glacier recession before ~2008 is higher than in the period after. Could this be an effect of changing ice flow velocities or thicknesses? An effect of subglacial topography? Warmer air, thus warmer water temperatures? Stronger winds creating thermal-erosional notches at the glacier terminus? More precipitation, and hence higher water levels? Please discuss, possibly with available data (not) correlating with your observations.

L249-278: These two paragraphs could be shortened given that the 'trend in length is similar to that of the glacier area', as the authors write. Readers can infer the main messages from Fig 3.

L293-302: If it is a hypothesis that glacier retreat occurs synchronously along the glacier front, then the authors should clearly emphasize this hypothesis, and test it. In this regard, I fail to see any ice avalanches (I guess the authors mean glacier calving) in Fig. 4 and Fig. 5, which the authors claim to be the main cause of the asynchronous glacier retreat. Neither do I see whether these avalanches are 'huge' or 'small', and whether these occurred as one single, or as a sequence of many smaller events. In this regard, it remains difficult to judge whether the three proposed categories (L303-308) have a more general implication between glacier retreat and lake expansion. Just as an idea: could it be that the total amount of calved glacier mass is a consequence of how fractured the glacier is at the glacier terminus? Could the authors measure the density of seracs at the tongue?

L322: How 'huge' needs a wave to be to 'put pressure on the moraine dams and increase the failure risk'? What is the minimum size of an ice avalanche and what is the amount of water to be displaced in order to overtop the dam?

L323-325: Please clarify what 'insignificant' and 'not obviously' mean.

L329: Please clarify what 'significantly later' means.

L330-331: Please reformulate and fill this sentence with tangible content.

L335: How do the authors calculate this 'trend'? What about the errors in the parameters of intercept and slope, and the goodness of fit of the model?

L336-337: Not sure how this sentence follows from the previous.

L347-354: Are the 6-years GoLIVE data representative for mean flow velocities your entire study period, in the light of the pattern that the authors observed for the mean surface velocities?

L349-350: How can you say that 'the fastest ice flow occurred in summer (May, June, and July)' without having data for these months? This finding largely depends on the polynomial function used to model the velocities with missing data.

L366-375: Given that Longbasaba Glacier is in direct contact to its lake, a unit change of the glacier area or length must result in the same unit change of the lake area, regardless of the total mass loss at the glacier terminus. What is here the surprising result to report?

L381-384: Which figure shows that 'the proportions of the mass contributions from the lowering of the glacier surface decreased to approximate 50%'? Furthermore, the authors conclude 'that the mass contributions from the glacier motion have increased during these periods.' Do this mean that glaciers needed to have a simultaneous increase in flow velocities to maintain the loss in glacier mass? I cannot infer this from Fig. 6.

L415: Fig. 9a: Is it possible that assuming constant values for surface lowering before and after 2000 could largely bias the ratios in Fig. 9b? How confident are the authors that surface lowering is indeed constant over time? Where are the error bars for all data

points? Could the error bars overlap so that the trends, which the authors mention in the text, are just spurious?

L430-431: Not sure whether I can follow this conclusion. Assuming an average increase of the lake level by 0.8 m per year, I calculate a total increase in lake level of 24 m in 30 years. This would be a quarter of the maximum lake depths, and hence speculatively more than 1.5% of the total water volume.

L432-440: Without showing data for Longbasaba Glacier, nor data from previous studies, it is difficult to judge whether evaporation and infiltration are really negligible. Please discuss in more detail.

L441-442: How does Longbasaba Glacier rank among the mean lowering rate for the Himalayas?

L450: That's a question of chicken or egg. Do glaciers thin because of lake expansion or do lakes expand because glacier thinning? This is a paramount question of regarding glacier-and-lake feedbacks where many researchers would love to have an answer for. Please discuss in more detail.

L459-461: Please reformulate these sentences more clearly. Where and when are air temperatures highest (and how high)? Which region is sensitive to what, and how do you define significant here?

L467-468: Maurer et al. (2019) collated data from weather stations and found that "Regional studies of precipitation trends in the Himalayas do not suggest a widespread decrease in precipitation over the past four decades". How does their statement match with the statement from the authors?

L459-478: What is purpose of this paragraph with regards to Longbasaba Glacier? Sounds more like material for the introduction?

L480: What is so 'specific' of Longbasaba's response to climate change?

L497-498: I cannot follow the conclusion that debris cover is important without formal analysis. How can we distinguish, which share of the mass loss comes from the debris cover on top of glaciers?

L503: What are these 'topographic features' and are these also present at Longbasaba lake?

L510-512: The authors could consider a simple correlation to test whether there is indeed no 'obvious relationship' among these variables.

L513: Where do the authors infer that the glacier area 'experienced continuous and accelerating decrease'? Not sure whether I can infer this finding from Fig. 3.

L517-518: Please elaborate more clearly in which regard these trends (are they robust?) are contrasting?

L519-520: Is the situation so 'complicated', 'diverse', and 'complex' that the amount of data generated in this study offers no further explanation on how glaciers and lakes interact? Somehow undermines the main scope of this manuscript.

L525-543: Seems more like material for the introduction. How do these paragraphs relate to the findings in this study?

L538: Do the authors mean 'air temperatures' when talking about climate change?

L541: 'specific climate changes and their influence on hazards': Which changes? Influence in which direction? Which hazards? And again, what could be a solution for these 'complex interactions'? What do we learn from this study?

L542-543: Not sure whether the selected references are appropriate for this statement.

L545-546: Do ice avalanches occur more frequently? Or how do they increase the potential for GLOFs?

L551: Is a rise in lake level of 6-11 m still negligible, as written in L421? What is the

actual rise in lake level?

L552, L555: How thick is the debris cover at the glacier terminus?

L558: How much is 'slight'?

L558-559: Not sure whether I can follow this conclusion: if ice avalanches have happened so frequently in the past, but still fail to trigger outburst floods, how can they still play 'a predominant role in the outburst risk'?

L560: All a matter of magnitude and frequency, and historical observations: How 'negligible' is the hazard from one rockslide that has a volume and/ or velocity 3, 10, 20 times larger than an average ice avalanche?

L563: Veh et al. (2019) report an unchanged frequency of outburst floods from moraine-dammed lakes.

564-567: The Himalayas host >20,000 glacier lakes (Maharjan et al., 2018), how many of these have been stabilised?

L571: Why has the possibility declined if ice avalanches are assumed to maintain occurring?

L572-573: In which way does Longbasaba Lake lag behind climate change? Is lake growth not an indicator for its response to atmospheric warming and glacier retreat?

L575-590: The conclusions are largely a repetition of the abstract. What have we learned from this study in conjunction with previous studies? What are the current limitations and what is the most important topic for future research?

Technical comments:

L34: Other studies have used the term 'lake-terminating'. Please check.

L35: 'tongues'

L35 (and all other locations in the manuscript): Please avoid using forward slashes.

[Figure]

Please use 'and' or 'or' instead.

L61: 'is in contact to' instead of 'is contacted by'?

L65-66: 'ice floe masses': suggest using 'floes' or 'icebergs'

L70: '5499 m': add 'a.s.l.' or 'above sea level'.

L72: Delete 'in 2009'

L73 and L80: 'terminus' instead of 'terminal'?

L76 (Fig. 1): Label inset in (a) with a letter.

L88: Please add the projection (UTM Zone XX)

L94: 'with minimum' instead of 'without'?

L95: 'suitable' instead of 'perfect'?

L99-101: Not necessary whether this table is necessary in the main text.

L103: 'center flowline' instead of 'main flowline'?

L103-105: Please reformulate this sentence.

L108: 'These data were extracted'. . .

L112: Why 'nevertheless'?

L115: delete 'described above'

L166, L191, L194, L206: Please correct the sub- and superscripts for h

L171-173: Please describe all terms (including v, h, and S) in the figure caption.

L185: What is MG?

L187: 'conversion' instead of 'conversation'

L193: Do you mean 'the number of pixels' instead of 'the pixel number'?

L226: 'following' instead of 'flowing' (or delete entirely)

L228: What is eac?

L251: Please add 'on average'

L283-284: A mass budget cannot be released.

L330: fluctuation = variations. Please consider one of these.

L373: fluctuation = variations = changes. Please consider one of these. Similar issue at L511, L513.

L377: 'was dominated' instead of 'was predominantly contributed'?

L414 (Fig. 9): 'surface lowering' instead of 'surface lowing'?

L443: 'combining' instead of 'combing'?

L507: 'Quincey'

L510: 'Dehecq'

References

Bhambri, R., Hewitt, K., Kawishwar, P., Kumar, A., Verma, A., Snehmani, Tiwari, S. and Misra, A.: Ice-dams, outburst floods, and movement heterogeneity of glaciers, Karakoram, Glob. Planet. Change, 180, 100–116, doi:10.1016/j.gloplacha.2019.05.004, 2019.

Brun, F., Berthier, E., Wagnon, P., Kääb, A. and Treichler, D.: A spatially resolved estimate of High Mountain Asia glacier mass balances from 2000 to 2016, Nat. Geosci., 10(9), 668–673, doi:10.1038/ngeo2999, 2017.

Dehecq, A., Gourmelen, N., Gardner, A. S., Brun, F., Goldberg, D., Nienow, P. W., Berthier, E., Vincent, C., Wagnon, P. and Trouvé, E.: Twenty-first century glacier slowdown driven by mass loss in High Mountain Asia, Nat. Geosci., 12(1), 22–27,

doi:10.1038/s41561-018-0271-9, 2019.

Farinotti, D., Huss, M., Fürst, J. J., Landmann, J., Machguth, H., Maussion, F. and Pandit, A.: A consensus estimate for the ice thickness distribution of all glaciers on Earth, Nat. Geosci., 12(3), 168–173, doi:10.1038/s41561-019-0300-3, 2019.

Haritashya, U., Kargel, J., Shugar, D., Leonard, G., Strattman, K., Watson, C., Shean, D., Harrison, S., Mandli, K. and Regmi, D.: Evolution and Controls of Large Glacial Lakes in the Nepal Himalaya, Remote Sens., 10(5), 798, doi:10.3390/rs10050798, 2018.

Khanal, N. R., Mool, P. K., Shrestha, A. B., Rasul, G., Ghimire, P. K., Shrestha, R. B. and Joshi, S. P.: A comprehensive approach and methods for glacial lake outburst flood risk assessment, with examples from Nepal and the transboundary area, Int. J. Water Resour. Dev., 31(2), 219–237, doi:10.1080/07900627.2014.994116, 2015.

Linsbauer, A., Frey, H., Haeberli, W., Machguth, H., Azam, M. F. and Allen, S.: Modelling glacier-bed overdeepenings and possible future lakes for the glaciers in the Himalaya—Karakoram region, Ann. Glaciol., 57(71), 119–130, doi:10.3189/2016AoG71A627, 2016.

Maharjan, S. B., Mool, P., Lizong, W., Xiao, G., Shrestha, F., Shrestha, R., Khanal, N., Bajracharya, S., Joshi, S. and Shai, S.: The status of glacial lakes in the Hindu Kush Himalaya-ICIMOD Research Report 2018/1 (2018)., ICIMOD Res. Rep., (2018/1), 2018.

Maurer, J. M., Schaefer, J. M., Rupper, S. and Corley, A.: Acceleration of ice loss across the Himalayas over the past 40 years, Sci. Adv., 5(6), eaav7266, doi:10.1126/sciadv.aav7266, 2019.

Nie, Y.: A regional-scale assessment of Himalayan glacial lake changes using satellite observations from 1990 to 2015, Remote Sens. Environ., 13, 2017.

Nie, Y., Liu, Q., Wang, J., Zhang, Y., Sheng, Y. and Liu, S.: An inventory of

historical glacial lake outburst floods in the Himalayas based on remote sensing observations and geomorphological analysis, Geomorphology, 308, 91–106, doi:10.1016/j.geomorph.2018.02.002, 2018.

Quincey, D. J., Richardson, S. D., Luckman, A., Lucas, R. M., Reynolds, J. M., Hambrey, M. J. and Glasser, N. F.: Early recognition of glacial lake hazards in the Himalaya using remote sensing datasets, Glob. Planet. Change, 56(1–2), 137–152, doi:10.1016/j.gloplacha.2006.07.013, 2007.

Richardson, S. D. and Reynolds, J. M.: An overview of glacial hazards in the Himalayas, Quat. Int., 65–66, 31–47, doi:10.1016/S1040-6182(99)00035-X, 2000.

Rounce, D. R., McKinney, D. C., Lala, J. M., Byers, A. C. and Watson, C. S.: A new remote hazard and risk assessment framework for glacial lakes in theNepal Himalaya, Hydrol. Earth Syst. Sci., 20(9), 3455–3475, doi:10.5194/hess-20-3455-2016, 2016.

Sharma, R. K., Pradhan, P., Sharma, N. P. and Shrestha, D. G.: Remote sensing and in situ-based assessment of rapidly growing South Lhonak glacial lake in eastern Himalaya, India, Nat. Hazards, 93(1), 393–409, doi:10.1007/s11069-018-3305-0, 2018.

Somos-Valenzuela, M. A., McKinney, D. C., Rounce, D. R. and Byers, A. C.: Changes in Imja Tsho in the Mount Everest region of Nepal, The Cryosphere, 8(5), 1661–1671, doi:10.5194/tc-8-1661-2014, 2014.

Song, C., Sheng, Y., Ke, L., Nie, Y. and Wang, J.: Glacial lake evolution in the southeastern Tibetan Plateau and the cause of rapid expansion of proglacial lakes linked to glacial-hydrogeomorphic processes, J. Hydrol., 540, 504–514, doi:10.1016/j.jhydrol.2016.06.054, 2016.

Song, C., Sheng, Y., Wang, J., Ke, L., Madson, A. and Nie, Y.: Heterogeneous glacial lake changes and links of lake expansions to the rapid thinning of adjacent glacier termini in the Himalayas, Geomorphology, 280, 30–38, doi:10.1016/j.geomorph.2016.12.002, 2017.

Tsutaki, S., Fujita, K., Nuimura, T., Sakai, A., Sugiyama, S., Komori, J. and Tshering, P.: Contrasting thinning patterns between lake- and land-terminating glaciers in the Bhutanese Himalaya, The Cryosphere, 13(10), 2733–2750, doi:10.5194/tc-13-2733-2019, 2019.

Veh, G., Korup, O., Roessner, S. and Walz, A.: Detecting Himalayan glacial lake outburst floods from Landsat time series, Remote Sens. Environ., 207, 84–97, doi:10.1016/j.rse.2017.12.025, 2018.

Veh, G., Korup, O., von Specht, S., Roessner, S. and Walz, A.: Unchanged frequency of moraine-dammed glacial lake outburst floods in the Himalaya, Nat. Clim. Change, 9(5), 379–383, doi:10.1038/s41558-019-0437-5, 2019.

Wang, S., Qin, D. and Xiao, C.: Moraine-dammed lake distribution and outburst flood risk in the Chinese Himalaya, J. Glaciol., 61(225), 115–126, doi:10.3189/2015JoG14J097, 2015a.

Wang, W., Xiang, Y., Gao, Y., Lu, A. and Yao, T.: Rapid expansion of glacial lakes caused by climate and glacier retreat in the Central Himalayas, Hydrol. Process., 29(6), 859–874, doi:10.1002/hyp.10199, 2015b.

Wang, W., Gao, Y., Iribarren Anacona, P., Lei, Y., Xiang, Y., Zhang, G., Li, S. and Lu, A.: Integrated hazard assessment of Cirenmaco glacial lake in Zhangzangbo valley, Central Himalayas, Geomorphology, 306, 292–305, doi:10.1016/j.geomorph.2015.08.013, 2018.

Watanabe, T., Lamsal, D. and Ives, J. D.: Evaluating the growth characteristics of a glacial lake and its degree of danger of outburst flooding: Imja Glacier, Khumbu Himal, Nepal, Nor. Geogr. Tidsskr., 14, 2009.

Xin, W., Shiyin, L., Wanqin, G. and Junli, X.: Assessment and Simulation of Glacier Lake Outburst Floods for Longbasaba and Pida Lakes, China, Mt. Res. Dev., 28(3/4), 310–317, doi:10.1659/mrd.0894, 2008.

---

## Referee Comment (RC2) · Anonymous Referee #2 · 7 Apr 2020

The manuscript focuses on a peri-glacial lake expansion of Langbasaba Glacier in the Central Himalaya. The topic is relevant to TC but manuscript is has some major flaws and is not ready for publication in TC. My decision is "Reject". The language is sloppy and many times difficult to follow.

Authors have estimated the contribution of glacier surface lowering, snout retreat and changes in glacier velocity to the glacier lake volume. All the data to derive these estimates are from gridded sources, no comparison or validation is done using field data (surface lowering or surface velocities). In this situation, it is difficult to constrain the actual uncertainties in their estimates. Authors assumed evaporation or sublima-

tion processes to be negligible. In the central Himalaya, evaporation/sublimation were estimated to be quite high (up to 21% of annual snowfall) (Stigter et al., 2018). Ignoring sublimation would lead an overestimation of the mass contributions of the glacier changes to the lake water volume, as highlighted by the authors. Authors, estimated the water volumes coming from different sources but did not discuss how much discharge is generated from the lake. They should have discussed the complete water cycle of the lake. The discussion part (section 5.2 and 5.3) reads like literature review and could not bring any new science based on their results. Author should discuss the key questions: 1) what is the threshold water volume (capacity of moraine dam) Longbasaba lake can hold, 2) is this lake potentially dangerous (if there is any habitat downstream), 3) when the lake may burst if the lake expansion rate continues in similar fashion, and possible remedies to control the GLOF (if there is downstream habitat that can be affected).

Some minor suggestions: L 21: replace "lowing" with "lowering"

L 25: "Due to the areal expansion, decreasing mass contributions from parent glacier shrinkage, and some mitigation measures by local governments to improve the drainage systems, the potential risk of outburst for Longbasaba Lake has continuously decreased during the last decade." I could not see any mitigation measure from government on this glacier discussed in this manuscript. Further, I don't understand how authors concluded that the decreasing mass contribution from glacier led to decreased risk over the last decade. In any case, the lake volume is continuously increasing so as the risk.

L 34: "lake-terminating"

L 51: "downstream communities"

L 66: what is "floe masses"?

L 77: show the lake stream in figure 1A.

L 131: "...ice flow and were...". Remove "and".

L 239: "the estimated accuracy..."

L 237-243: How the uncertainty in surface velocity was estimated?

L 250-252: Rephrase the sentence. Not clear.

L 255: "decreased dramatically"

L 259: what are those other periods? Describe here.

L 263: which period?

L 268: it is confusing to see length changes in % a-1, please give the changes in meter. % a-1 is mostly used for areal changes.

L 279: which period, glacier showed reduced mass wastage?

L 280: are these glacier-wide elevation changes?

L 293: I would suggest to use "sides" than "flanks"

L 327-329: The sentence is not clear, please rephrase.

L 373: "...fluctuation in the variations in the changes in the glacier area." Not clear to me. Please rephrase.

L 421-422: reference for the assumption?

L 433: "infiltration in the ground' would lead to underestimation in mass contributions of glacier changes. Please check.

L 489: it is Banerjee and Shankar, 2013.

Reference: Stigter, E. E. et al. 2018. The importance of snow sublimation on a Himalayan glacier. Front. Earth Sci. 6, 108.

[Figure]

Please also note the supplement to this comment:
https://www.the-cryosphere-discuss.net/tc-2019-259/tc-2019-259-RC2-
supplement.pdf
* * *